# Artificial Intelligence-Based Hyper Accuracy Three-Dimensional (HA3D^®^) Models in Surgical Planning of Challenging Robotic Nephron-Sparing Surgery: A Case Report and Snapshot of the State-of-the-Art with Possible Future Implications

**DOI:** 10.3390/diagnostics13142320

**Published:** 2023-07-10

**Authors:** Michele Di Dio, Simona Barbuto, Claudio Bisegna, Andrea Bellin, Mario Boccia, Daniele Amparore, Paolo Verri, Giovanni Busacca, Michele Sica, Sabrina De Cillis, Federico Piramide, Vincenzo Zaccone, Alberto Piana, Stefano Alba, Gabriele Volpi, Cristian Fiori, Francesco Porpiglia, Enrico Checcucci

**Affiliations:** 1Division of Urology, Department of Surgery, SS Annunziata Hospital, 87100 Cosenza, Italy; claudio.bisegna@gmail.com (C.B.); v.zaccone@aocs.it (V.Z.); 2Department of Oncology, Division of Urology, San Luigi Gonzaga Hospital, University of Turin, 10043 Orbassano, Italy; simona.barbuto@medics3d.com (S.B.); danieleamparore@hotmail.it (D.A.); paoloverri05@gmail.com (P.V.); giovannibusacca95@gmail.com (G.B.); michelesica1991@gmail.com (M.S.); sabrinatittidecillis@gmail.com (S.D.C.); federico.piramide@gmail.com (F.P.); alb.piana@gmail.com (A.P.); cristian.fiori@unito.it (C.F.); porpiglia@libero.it (F.P.); enrico.checcucci@ircc.it (E.C.); 3Romolo Hospital, 88821 Rocca di Neto, Italy; stefanoalba78@gmail.com; 4Department of Surgery, Candiolo Cancer Institute, FPO-IRCCS, 10060 Candiolo, Italy; volpi_gabriele@yahoo.it

**Keywords:** kidney cancer, 3D models, robotics, three-dimensional, artificial intelligence

## Abstract

Recently, 3D models (3DM) gained popularity in urology, especially in nephron-sparing interventions (NSI). Up to now, the application of artificial intelligence (AI) techniques alone does not allow us to obtain a 3DM adequate to plan a robot-assisted partial nephrectomy (RAPN). Integration of AI with computer vision algorithms seems promising as it allows to speed up the process. Herein, we present a 3DM realized with the integration of AI and a computer vision approach (CVA), displaying the utility of AI-based Hyper Accuracy Three-dimensional (HA3D^®^) models in preoperative planning and intraoperative decision-making process of challenging robotic NSI. A 54-year-old Caucasian female with no past medical history was referred to the urologist for incidental detection of the right renal mass. Preoperative contrast-enhanced abdominal CT confirmed a 35 × 25 mm lesion on the anterior surface of the upper pole (PADUA 7), with no signs of distant metastasis. CT images in DICOM format were processed to obtain a HA3D^®^ model. RAPN was performed using Da Vinci Xi surgical system in a three-arm configuration. The enucleation strategy was achieved after selective clamping of the tumor-feeding artery. Overall operative time was 85 min (14 min of warm ischemia time). No intra-, peri- and post-operative complications were recorded. Histopathological examination revealed a ccRCC (stage pT1aNxMx). AI is breaking new ground in medical image analysis panorama, with enormous potential in organ/tissue classification and segmentation, thus obtaining 3DM automatically and repetitively. Realized with the integration of AI and CVA, the results of our 3DM were accurate as demonstrated during NSI, proving the potentialities of this approach for HA3D^®^ models’ reconstruction.

## 1. Introduction

In recent times, there has been a significant shift in urologic surgery towards personalized patient care, particularly in cases involving malignancies. When it comes to renal cancer and its associated surgical procedures, striking a balance between ensuring oncological safety and maximizing functional recovery is of paramount importance. As a result, surgeons are increasingly employing nephron-sparing techniques to manage renal lesions, even in complex tumor cases [1,2].

In order to achieve this objective, the utilization of image-guided interventions plays a crucial role in optimizing the customization of surgical procedures. By employing preoperative imaging techniques such as computer tomography scans, a three-dimensional (3D) reconstruction of the kidney and its lesions can be obtained [3].

The foundation for creating these 3D models lies in image acquisition. The most reliable method for reconstructing kidney tumors and vessels is contrast-enhanced CT scans. In the past, 3D models were generated automatically through radiological software. However, modern advancements have led to the development of dedicated technologies that enable highly accurate and precise 3D virtual reconstructions known as hyper accurate 3D (HA3D) models. These HA3D models are refined through collaborative efforts between radiologists, urologists, and engineers, utilizing professional software to enhance accuracy and ensure the fidelity of the final product. If the model meets specific technical requirements, it can be considered a fully functional medical device [4]. In recent years, there has been a significant and rapid evolution in the development of 3D models. These models have progressed beyond representing the patient’s anatomy as highly detailed static images and now incorporate perfusion area information [5,6]. By applying mathematical models, it becomes possible to predict the specific area of the kidney parenchyma supplied by each arterial branch. These enhanced 3D models enable a more precise approach to selective clamping during surgery. Instead of relying on empirical assumptions about the arteries supplying the tumor, surgeons can now make informed decisions based on mathematical demonstrations of the perfusion areas. This represents a shift in perspective. Rather than considering the direction of the artery towards the tumor, the focus is on understanding the area of tumor growth and which arteries supply it. This advancement in selective clamping techniques allows for a more tailored and accurate approach, improving surgical outcomes and minimizing unnecessary damage to healthy tissue. It signifies a significant improvement in the field, leveraging mathematical models and enhanced 3D models to guide surgical interventions based on the perfusion characteristics of the tumor.

Once these 3D models are created, they can be employed during surgery to assist in a cognitive or augmented reality (AR) capacity [7,8].

The potential and desirability of 3D guidance in nephron-sparing surgery have been previously demonstrated. However, only recently, a systematic review and meta-analysis published last year provided evidence that 3D model guidance during minimally invasive partial nephrectomy is associated with certain surgical preferences. These include a higher utilization of selective clamping instead of total clamping, a preference for tumor enucleation over resection, and a reduced likelihood of opening the collecting system [9].

Despite the promising outcomes of 3D model-guided surgery, there are technical limitations that present obstacles requiring resolution. For instance, in the context of augmented reality (AR), achieving precise real-time tracking remains an ongoing challenge. Additionally, tissue deformation poses another significant obstacle, as the application of realistic biomechanical models in clinical practice is still under development. Lastly, the creation of 3D models is a time-consuming process that necessitates the involvement of trained engineers.

To address these challenges, the application of artificial intelligence (AI) algorithms can be instrumental in overcoming some of the limitations associated with human barriers [10]. AI is defined as the capability of machines to perform tasks and solve problems without explicit programming. Various methods exist for medical image segmentation, including traditional approaches based on regions and edges, as well as deep learning-based methods. However, traditional methods have limitations due to non-uniform grayscale characteristics, individual variations, image artifacts, and noise. Achieving accurate heart segmentation, for instance, is challenging using these methods. In contrast, deep learning models have shown promising results in image segmentation, improving the accuracy of disease diagnosis and reducing irrelevant computations.

In this context, we present a case where a 3D model reconstruction was achieved using artificial intelligence algorithms, demonstrating the fidelity of the reconstructed 3D model during the intraoperative phase.

## 2. Case Presentation

A 54-year-old Caucasian female with no past medical or surgical history was referred by her primary care physician to the urologist for an incidental detection of a right renal mass at an abdominal ultrasound scan, performed to investigate a recently appeared dyspepsia. The patient did not have a family history of urogenital cancers, nor alcohol, tobacco, or other recreational substances consumption. A contrast-enhanced abdominal CT scan confirmed the presence of a right-sided renal tumor, with no signs of distant metastasis. No local or systemic symptoms related to the neoplasm were present at the moment of the diagnosis. At objective assessment, no palpable flank masses or other significant findings were found. The complete blood count and biochemical profile of the patient were within reference ranges, with a conserved renal function (i.e., preoperative glomerular filtration rate [eGFR] 118 mL/min/1.73 m^2^, serum creatinine of 0.79 mg/dL). No alterations were recorded in urinalysis and no macroscopic hematuria was reported.

### 2.1. Nephrometric Characteristics of Renal Cancer

A preoperative contrast-enhanced abdominal CT scan (1.5 mm thickness) confirmed a 35 × 25 mm renal lesion located on the anterior surface of the upper pole of the right kidney (Figure 1).

Two main renal arteries were detected, with the superior one supplying the region affected by the neoplasm and the inferior one directed to the lower pole. Three venous branches tributary to the inferior vena cava (IVC) were found, with an additional early branch of the first cranial one directed to the lesion. According to preoperative aspects and dimensions used for an anatomical (PADUA) nephrometry classification, it was a “low complexity” renal tumor (PADUA score of 7: 1 point for “tumor size”; 1 point for “exophytic” rate; 1 point for “collecting system” involvement; 1 point for “sinus” involvement; 1 point for “renal rim”; and 2 points for polar location). The clinical stage was cT1a cN0 cM0. It is noteworthy to say that in the present case, nephrometric characteristics were preoperatively assessed via HA3D reconstruction and bi-dimensional (2D) CT images, maintaining the same risk/complexity category. Notably, the adoption of HA3D models with AI may help the surgeon assess the surgical complexity of challenging NSI and patient-specific anatomic understanding. Potentially overcoming the abstraction process needed when consulting 2D images, HA3D models may indeed increase the accuracy of applying nephrometry scores, best predicting postoperative complications [11]. 

### 2.2. 3D Reconstruction with AI

The CT scan images in DICOM format were processed by Medics Srl (https://www.medics3d.com/, accessed on 7 June 2023, Turin, Italy) to obtain a Hyper Accuracy Three-dimensional (HA3D^®^) model, which is a patient-specific 3D reconstruction of both the renal tumor and the intraparenchymal structures, built to improve the accuracy and the efficacy of the preoperative planning for a clinical case of RAPN. 

The operative workflow to obtain a HA3D^®^ model consists of three consecutive steps: the study of medical images, segmentation, and post processing (Figure 2).

The first step is the evaluation of the quality of multiphase CT scan images by bioengineers. Then, each specific anatomical region of interest (ROI) is reconstructed during the segmentation process by combining the use of different software with proprietary algorithms in close cooperation with bioengineers, urologists, and radiologists. The collecting system is segmented from the excretory phase while the renal veins, renal arteries, renal parenchyma, tumor, and cysts are segmented, taking images from the arterial phase. Finally, after appropriate post processing through semi-automatic proprietary algorithms based on computer vision, the HA3D^®^ virtual model is obtained (Figure 3).

Artificial intelligence (AI) is breaking new ground in the field of medical image analysis, such as CT scans and magnetic resonance imaging (MRI). Deep learning is a branch of AI that has shown enormous potential in this area and that can be applied to classify and segment organs and tissues and, thus, obtain 3D models automatically and repetitively.

In this case study, a mutual combination of computer vision and deep learning segmentation tools was used.

Renal arteries and veins, collecting system, tumors, and cysts were segmented through Mimics Medical 21.0 (Materialise, Leuven, Belgium) using automatic and manual segmentation techniques that use computer vision algorithms based on grayscale, according to the previous description, as the gold standard [12]. A part of the manual refining process is necessary to better define the contours of the anatomical parts and remove “noise pixels”. With good-quality images, it is possible to identify at least up to the third intraparenchymal branch, giving the model an outstanding level of accuracy.

Renal parenchyma, aorta, and inferior vena cava were segmented using an automatic tool based on AI software. The TotalSegmentator tool [13] by 3D Slicer (https://www.slicer.org/, accessed on 7 June 2023) currently represents the gold standard in the artificial intelligence segmentation process. 

It takes as input any CT scan series and it can segment in automatic the anatomical structures of interest, such as abdominal organs, muscles, bones, and large vessels (Figure 4).

Since this tool merges the parenchyma and the exophytic part of the tumor without separating the cortex from the medulla, in order to obtain a separation of the cortex from the medulla, a Boolean subtraction is employed using a specific proprietary algorithm. 

The AI segmentation results are accurate enough for most purposes, even if there can be a couple of millimetric segmentation errors.

To better plan the clamping strategy, the 3D model was subsequently implemented by Medics bioengineers with an internal proprietary algorithm based on the Voronoi diagram, to visualize the perfusion regions for each segmental artery entering the renal parenchyma [5,14]. The perfusion regions were then calculated by subdividing the healthy parenchyma into anatomic volumes (i.e., voxel, the 3D equivalent of a 2D pixel), according to the morphology and proximity of each arterial branch. 

### 2.3. Surgical Technique for Robot-Assisted Partial Nephrectomy (RAPN)

The surgery was performed by a surgeon with extensive experience in the field of nephron-sparing surgery, who performed more than 350 partial nephrectomies, using the Da Vinci Xi surgical system (Intuitive Surgical, Sunnyvale, CA, USA) with a 30° lens (exploiting the potentialities of 30° up-down position) in a three-arm configuration. The patient was positioned in a left flank decubitus with the surgical table mildly flexed and positioned in a slight Trendelenburg position to better expose the costovertebral angle. Pneumoperitoneum was induced using a Verres needle placed laterally to the rectus muscle across from the 12th rib. Once the pneumoperitoneum was achieved, the optical trocar was inserted, and the peritoneal cavity was inspected for potential injuries or adhesions. A total of three robotic 8 mm ports were placed under direct vision in a linear fashion, at the lateral border of the rectus muscle. Two more trocars for the assistant were used, one of 5 mm and another of 12 mm for the Air Seal system, keeping a constant pressure of 12 mmHg during the surgery. An additional 5 mm subcostal port was placed to retract the liver using a locking Allis clamp. The white line of Toldt was then released and the right colon medially retracted, exposing Gerota’s fascia. After its dissection, the renal hilum was approached, identifying the double renal arterial supply and the multiple veins. Then, renal parenchyma was released from the adherent surrounding adipose tissue, up to the upper pole. After gaining more space by dissecting the tissues, it was possible to easily identify the tumor lesion, constantly aided by the HA3D^®^ virtual model. The anatomical limits of renal mass were additionally confirmed using intraoperative ultrasound, thus delineating its contours. After the renal pedicle isolation, thanks to the patient-specific HA3D^®^ model with perfusion algorithms, a selective clamping of the cranial renal arterial branch, responsible for most of the arterial support of the tumor lesion according to the 3D model information, was planned. Ischemic areas were confirmed with near-infrared fluorescence imaging (NIRF) using indocyanine green fluorescence (IGF), showing that only 33% of renal parenchyma was undergoing selective warm ischemia. Therefore, an enucleative resection strategy starting from the enucleation plane was performed. Through the use of two robotic arms, generating gentle tractions to better expose the resection bed, the dissection plane was developed between the so-called peritumoral “pseudocapsule” and the surrounding healthy parenchyma. To reach perfect hemostasis, a single-layer cortical renorrhaphy was performed using interrupted 2-0 (26 mm needle) polyfilament sutures placed at intervals of 1 cm using the sliding-clip technique with Hem-o-Lok^®^ clips. In addition, Floseal^®^ (Baxter Healthcare Corporation Fremont, Hayward, CA, USA) hemostatic matrix was applied to further decrease the chance of postoperative bleeding. Of note, the surgeon benefited from the support of the HA3D^®^ virtual model for intraoperative navigation during all the prominent phases of the surgery. Particularly, this allowed us to adapt and refine the decision-making process, leading to the effective application of the preoperative selective clamping and enucleative resection strategy. Digital support also allowed a constructive discussion about the clinical case with the rest of the team, proving to be very useful in encouraging the surgeon to share his choices made during the procedure. 

The total operative time at the robotic console for this RAPN case was 85 min, including the warm ischemia time (WIT) of 14 min. Estimated blood loss was 120 mL. No intraoperative complications were recorded and the postoperative stay was overall uneventful. Urinary catheter removal occurred on postoperative day (POD) 1, whilst the surgical drain was removed on POD 3, after intestinal canalization recovery. Patient discharge was on POD 4 in good general condition, in the absence of major pain, with eGFR of 103 mL/min/1.73 m^2^ and serum creatinine of 0.88 mg/dL (Figure 5).

A dedicated uropathologist performed the histopathological specimen examination, diagnosing an ISUP nucleolar grade 2 (G2) clear cell renal cell carcinoma (ccRCC) of 40 × 35 mm, without evidence of necrosis or sarcomatoid features. No positive surgical margins were detected on the tumor sample, defined as the absence of tumor cells touching the outer inked surface of the specimen (pT1a Nx Mx R0).

## 3. Discussion

Recently, 3D models gained a wide diffusion in urological surgery, especially in nephron-sparing interventions, but also in other surgical procedures, such as robotic-assisted radical prostatectomy (RARP) [3].

The availability of 3D models improves the quality of preoperative planning, allowing us to display precisely the anatomical details of both the healthy tissues and the tumor. Furthermore, the vascularization is thoroughly represented, effectively aiding the surgeon during the intraoperative decision-making process; finally, the augmented reality images guide the surgeon in real time, during the surgical procedure, serving as a surgical navigation tool [5,9,15,16]. In this scenario, besides virtual reality technologies, 3D printed models have been recently developed and tested for effectiveness in improving preoperative surgical planning, also demonstrating their utility for training and surgical simulations, thus recreating the environment of real interventions but in safe conditions for the patients [17,18].

As recently reported in [4], the 3D reconstruction for preoperative planning should meet some essential clinical requirements to allow for effective support during the intervention.

In fact, the quality and reality of the models are key elements, and the anatomical structures should be clearly visualized in order to build a faithful representation of the real organ. In particular, focusing on RAPN, it is important to visualize the whole vascular tree (i.e., renal arteries and veins) to better understand which vessels directly supply the tumor; in addition, the study of the resection bed is essential to selectively manage the intraparenchymal structures during the reconstruction phase, with the final objective being to perform a super selective suture, minimizing the contribution of the suture itself on the ischemic damage. Finally, a correct representation of both the collecting system and calyxes allows us to analyze any areas of contact between these structures and the mass. 

Currently, the creation of high-quality 3D models is mainly demanded by dedicated bioengineers who, thanks to a strict and constant collaboration between urologists and radiologists, use dedicated software authorized for medical use and are able to obtain hyper accurate models.

However, this process is affected by numerous, time-consuming and repetitive tasks, which can ultimately lead to an increasing lack of precision, with the risk to create systemic errors.

In this setting, the advent of artificial intelligence plays a fundamental role and may represent the technology finally providing the engineers the solution to solve this issue [10]. 

Up to now, the application of artificial intelligence techniques alone does not allow us to obtain a 3D model useful for planning a RAPN, as the minimal clinical requirements are not met. Renal arteries and veins are not automatically reconstructed; therefore, it is not possible to describe the renal pedicle to study the resection bed and to identify which vessels supply the tumor. The tumor, joined to the renal parenchyma, is not reproduced in its endophytic part, and, therefore, it is not possible to determine its extension and morphology. Finally, the collecting system is absent (Figure 6).

The integration of artificial intelligence with computer vision algorithms seems promising as it allows us to speed up the process and to get results equivalent to those obtained with the gold standard (manual) approach, with partial independence from the operator.

In the future, further improvement of deep learning algorithms with multiple datasets training them to recognize more details will allow us to achieve optimal results, meeting clinical requirements and realizing detailed 3D reconstructions using artificial intelligence only.

In addition, deep learning will potentially be inserted in all the phases of the process, ensuring a repeatable and operator-independent quality standard.

However, we strongly believe that, in current times, the results produced by artificial intelligence should always be verified by an expert user to validate the reliability and correctness of the results themselves.

## 4. Conclusions

In conclusion, herein we presented a case of a 3D model realized with the integration of artificial intelligence and a computer vision approach: the model’s results were accurate, proving the potential applicability of this approach for HA3D^®^ model reconstruction. In the future, the integration of different AI approaches will allow us to speed up the 3D model reconstruction process, maintaining the high quality of the models.

## Figures and Tables

**Figure 1 diagnostics-13-02320-f001:**
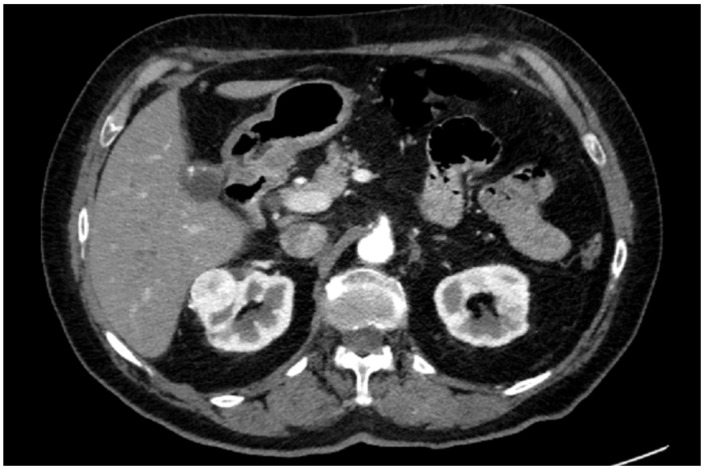
Axial slice of CT scan.

**Figure 2 diagnostics-13-02320-f002:**
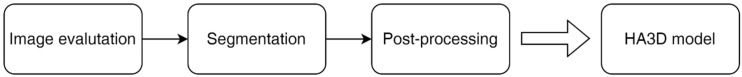
HA3D^®^ model workflow.

**Figure 3 diagnostics-13-02320-f003:**
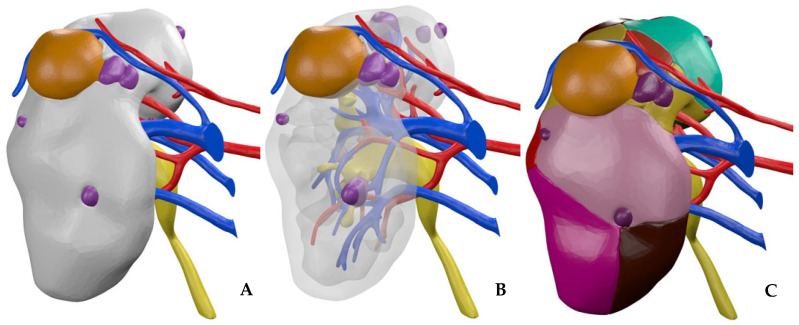
HA3D^®^ virtual model. (**A**) Standard 3D virtual model. (**B**) Standard 3D virtual model with transparent parenchyma. (**C**) A 3D virtual perfusion model.

**Figure 4 diagnostics-13-02320-f004:**
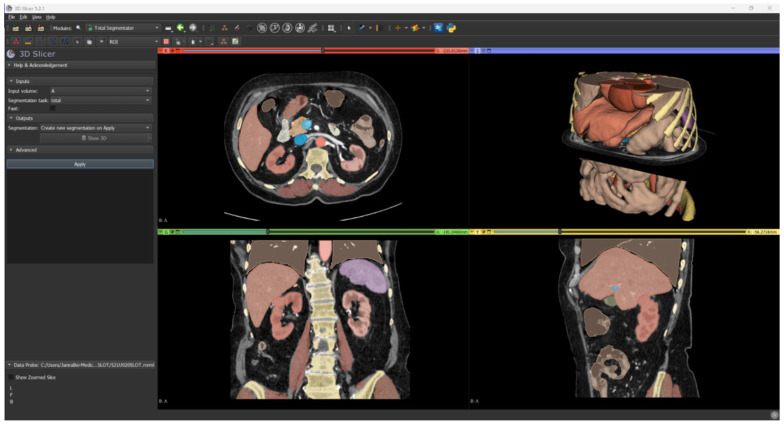
Result of automatic segmentation.

**Figure 5 diagnostics-13-02320-f005:**
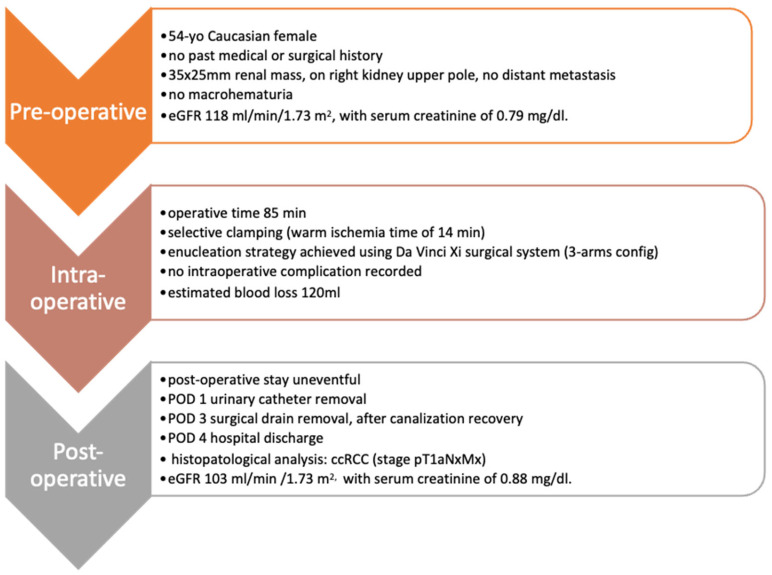
Pre-, peri- and post-operative patient characteristics.

**Figure 6 diagnostics-13-02320-f006:**
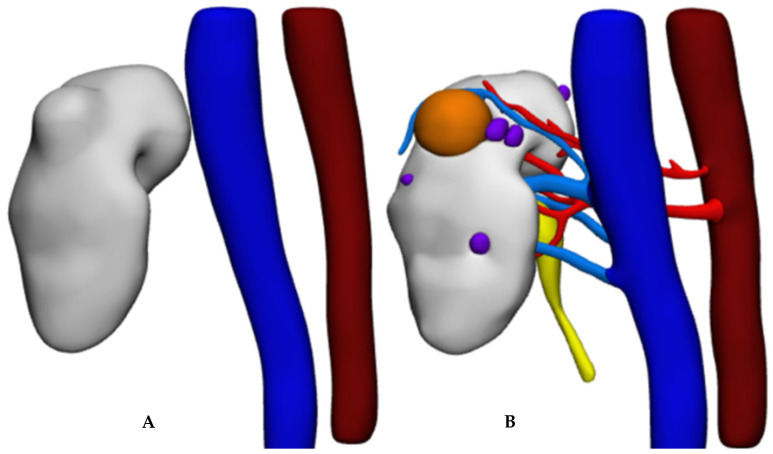
Results of 3D reconstruction: (**A**) deep learning only; (**B**) deep learning combined with computer vision.

## Data Availability

The data presented in this study are available on request from the corresponding author. The data are not publicly available due to privacy restrictions.

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
