# Peer review of "Artificial Intelligence-Based Hyper Accuracy Three-Dimensional (HA3D®) Models in Surgical Planning of Challenging Robotic Nephron-Sparing Surgery: A Case Report and Snapshot of the State-of-the-Art with Possible Future Implications"

_diagnostics, 2023, doi:10.3390/diagnostics13142320_

Round 1

Reviewer 1 Report

Di Dio et al. have written an interesting case report on the topic of applying artificial intelligence in clinical practice. I believe that the text is of high quality, with a suggestion for minimal corrections. The status of the surgical margins was not mentioned in the histopathological report, which I think is very important to mention it presents a case in which advanced technology was applied in the planning of a sparing operation. It would be nice to provide a comparison of the model obtained by artificial intelligence with the used nephrometric score.

Reviewer 2 Report

Thank you for giving me the opportunity to review this case report. In this case report authors aimed to present a case where a 3D model reconstruction was achieved using artificial intelligence algorithms, demonstrating the fidelity of the reconstructed 3D model during the intraoperative phase. Since AI and 3D modelling are of significance importance in surgical planning and training nowadays, I think this case worth’s considering for publication. However, I have some minor comments:

1.     
Figure 2 and 5, and also Table 1 might be removed.

2.     The way of utilizing 3D model during the surgery might be explained more. Did the surgeon view the 3D model many times during the surgery?

3.     Authors might also briefly discuss 3D printed phantom models in discussion section.

1.     Minor English editing is needed.

Round 2

Reviewer 2 Report

Please remove the aforementioned figures and table

Author Response

Point 1: Please remove the aforementioned figures and table

Response 1: Thank you again for suggesting. However we prefer to keep the aformentioned images and table, believing that they might be useful for the overall urderstanding of the article.  We still remain available to remove them if  editorial issues will be highlighted.